# Effect of Norelgestromin and Ethinylestradiol in Transdermal Patches on the Clinical Outcomes and Biochemical Parameters of COVID-19 Patients: A Clinical Trial Pilot Study

**DOI:** 10.3390/ph15060757

**Published:** 2022-06-17

**Authors:** Cortés-Algara Alfredo, Cárdenas-Rodríguez Noemí, Reyes-Long Samuel, Ortega-Cuellar Daniel, Ruz-Barros Rodrigo, Mondragón-Terán Paul, Escamilla-Tilch Mónica, Correa-Basurto José, Lara-Padilla Eleazar, Alfaro-Rodríguez Alfonso, Cortes-Altamirano José Luis, Bandala Cindy

**Affiliations:** 1Centro Médico Nacional 20 de Noviembre, ISSSTE, Mexico City 03100, Mexico; dr_cortes_a@hotmail.com (C.-A.A.); roemruba@gmail.com (R.-B.R.); p.mondragonteran@gmail.com (M.-T.P.); met171179@hotmail.com (E.-T.M.); 2Escuela Superior de Medicina, Instituto Politécnico Nacional, Mexico City 07738, Mexico; sam.long27@gmail.com (R.-L.S.); corrjose@gmail.com (C.-B.J.); vhlp1006@gmail.com (L.-P.E.); 3Laboratorio de Neurociencias, Instituto Nacional de Pediatría, Secretaría de Salud, Mexico City 04530, Mexico; noemicr2001@yahoo.com.mx; 4Laboratorio de Enfermedades Neurodegenerativas y Dolor, Instituto Nacional de Rehabilitación, Secretaría de Salud, Mexico City 06600, Mexico; alfa1360@yahoo.com.mx (A.-R.A.); jose_luiscortesalta@hotmail.com (C.-A.J.L.); 5Laboratorio de Nutrición Experimental, Instituto Nacional de Pediatría, Secretaría de Salud, Mexico City 04530, Mexico; dortegadan@gmail.com; 6Departamento de Investigación, Universidad Estatal del Valle de Ecatepec, Valle de Anahuac, Ecatepec 55210, Mexico State, Mexico

**Keywords:** COVID-19, norelgestromin, ethinylestradiol, estrogens

## Abstract

The disease caused by SARS-CoV-2 is still considered a global pandemic. Transdermal patches (TP) with immunoregulators such as estrogen and progesterone compounds could be a feasible option to treat COVID-19 because of their accessibility and relative safety. The objective of the current study was to evaluate the additional treatment with norelgestromin and ethinylestradiol in TP on the clinical and biochemical evolution of COVID-19 patients. The present is a clinical-trial pilot study that included subjects diagnosed with COVID-19, randomized into two groups; the experimental Evra^®^ TP (norelgestromin 6 mg and ethinylestradiol 0.60 mg) was administered such that it was applied on arrival and replaced at day 8 and day 15. The control continued with the conventional COVID-19 treatment protocol. A blood sample was taken each week in order to evaluate relevant biochemical parameters, clinical signs, and evolution. In total, 44 subjects participated in this study, 30 in the experimental group and 14 in the control group. Both groups were homogeneous in terms of age and comorbidities. The experimental group had a significantly lower hospital stay (*p* = 0.01), high flow supplemental oxygen (*p* = 0.001), mechanical ventilation (*p* = 0.003), and intubation (*p* = 0.01), and the oxygen saturation significantly increased (*p* = 0.01) in comparison with control group when patients were exposed to room air. A decrease in ferritin (*p* < 0.05) was observed, with no significant increase in ESR (*p* > 0.05), D dimer (*p* > 0.05) and platelets (*p* > 0.05) in an auto-controlled analysis in the experimental group. Norelgestromin and ethinylestradiol TP could be a safe and effective treatment for moderate and severe COVID-19 patients.

## 1. Introduction

Coronavirus disease 19 (COVID-19) is still considered as a global pandemic, caused by the severe acute respiratory syndrome-coronavirus type 2 (SARS-CoV-2), which triggered a health crisis that led to the deterioration of several sectors around the world. A large amount of research is now directed toward better understanding and treating COVID-19 [1,2,3]. SARS-CoV-2 enters the lungs through the angiotensin-converting enzyme 2 (ACE2) receptor, a member of the renin–angiotensin system (RAS), acting as a negative regulator in the homeostasis of RAS [4,5]. Recent evidence suggests that ACE2 interacts with SARS-CoV-2 through glycosylation sites that enhance its infectivity; thus, the glycosylation of the ACE2 protein increases the affinity of the virus S protein through several interactions, including glycan–glycan and glycan–protein, which facilitate the stability and affinity of viral binding to the target host receptor. Importantly, estrogen treatments may disrupt this interaction and block its entry into cells [6]; therefore, ACE2 is essential in the progression and clinical prognosis of COVID-19 [7,8]. Several studies have shown that sex is a biological variable that affects innate and adaptive immune responses and is evolutionarily conserved across diverse species [9]. Thus, chromosome genes and sex hormones, including estrogens, progesterone, and androgens, contribute to the differential regulation of immune responses between sexes [10,11]. COVID-19 presents different morbidity and mortality in males versus females. An analysis of data from European countries reveals that among all age groups older than 20 years, a greater mortality rate was found in men over women, whose fatality ratio ranged from 1.7–1.8 [12]; therefore, sex is an important risk factor in COVID-19 outcomes [13,14], as significant differences have been observed between males and females in terms of hospitalizations and deaths [15]. It has been shown that high levels of 17-β-estradiol and progesterone might improve immune response against COVID-19 [16], whereas estrogens can decrease the severity of infections by reducing chemokine and pro-inflammatory cytokine, including interferon γ (IFN-γ), tumor necrosis factor-α(TNF-α) and C-C-chemokine ligand-2 (CCL2) [17,18], whereas androgens increase COVID-19 infection by increasing blood neutrophils count and function, producing an increase in interleukin production (IL-1β, IL-10, IL-2), transforming growth factor-β (TGF-β) by immune cells, and reducing the antibody response to the infectious conditions [10]. In a similar fashion, androgen imbalance states are associated with COVID-19 complications in male patients, with no significant effect in females. In fact, a reduction in androgen signaling with five α-reductase inhibitors can reduce ACE2 levels and thereby decrease the internalization of the SARS-CoV-2, suggesting that androgen signaling inhibition can be a potential therapeutic strategy to reduce SARS-CoV-2 viral entry and mitigate severe manifestations in COVID-19 patients [19]. The evidence described above allowed us to hypothesize that female hormones, specifically estrogens, may play a role in disease onset. The norelgestromin and ethinylestradiol formula as a combined hormonal transdermal patch (Evra^®^) is a drug prescribed commonly as a contraceptive and hormone replacement therapy. Their predominant mechanism of action is the inhibition of ovulation by the suppression of gonadotropins [20]. Interestingly, ethinylestradiol can also modulate adaptive immunity through the regulation of T cells, cytokine production, and immune-related gene expression [21,22]; however, the mechanism of action of these hormones on immunity is largely unknown.

## 2. Results

### 2.1. Clinical and Demographic Data

The number of subjects that were recruited was 44; the experimental group was formed by 30 and the control group by 14. Table 1 shows the demographical data, comorbidities, and treatment features, these show that both groups were homogeneous in all, except for sex.

### 2.2. Follow-Up and Outcomes

#### 2.2.1. Principal Outcomes

Figure 1 shows the experimental group presented significantly fewer hospital stay days (18.5 ± 12.51 vs. 27.8 ± 6.67, *p* = 0.01); high-flow-oxygen supplementation days (16.5 ± 11.36 vs. 26.07 ± 6.77, *p* = 0.001); mechanical-ventilation days (2.27 ± 4.97 vs. 13.14 ± 13.1, *p* = 0.003); intubated days (1.53 ± 44.34 vs. 8.36 ± 12.83, *p* = 0.01); and oxygen saturation (covariate adjustment mean 88.33%, *p* = 0.01) in comparison with the control group.

Figure 2 shows the inflammation parameters. Ferritin decreases in the experimental group by approximately 50%, and considering a covariate adjustment mean of 1363.34 ng/dL, we found significant differences in the final measurements between the experimental and control group (*p* = 0.001). The experimental group did not show changes in the erythrocyte sedimentation rate (ESR) (*p* > 0.05), but in the control group, this parameter significantly increased (*p* = 0.0001). Considering a covariate adjustment mean of 31.72 mm for basal measurement in both groups, we found that the experimental group ESR was significantly decreased in comparison with the control group (36.6 ± 4.29 vs. 47.35 ± 4.44 mm, *p* = 0.01). C-reactive protein (CRP) and procalcitonin decreased in experimental group by 74% and 33%, respectively, but this was not significant (*p* > 0.05 in both parameters).

Figure 3 shows the blood count parameters; hemoglobin and hematocrit decreased in experimental group by 5.6 and 9.7%, respectively, but these differences were not significant (*p* > 0.05 in both). Cellular counting showed that erythrocyte and neutrophils did not significantly decrease (44% and 10%, respectively) in the experimental group, while leukocytes and lymphocytes showed a minimal increase (10.2% and 34.3%, respectively).

#### 2.2.2. Clinical Signs

Only the clinical signs were considered because in intubated subjects, it was not possible to assess clinical symptoms. Table 2 shows the frequencies of clinical signs at hospital arrival and in the final evaluation day in each group. A decrease of these signs is significant.

#### 2.2.3. Safety Outcomes

Estrogen compounds possess as a main secondary effect an increase in coagulation. Thus, we evaluated this risk through D dimer, fibrinogen, and platelet count (Table 2). Significant differences were not found when evaluating basal and final means of each group, except in fibrinogen in the experimental group, but this parameter decreased in the final measurement.

## 3. Discussion

The present study evaluated the administration of hormonal therapy in transdermal patches (Evra^®^, norelgestromin 6 mg and ethynyl estradiol 0.6 mg) in male and post-menopausal female subjects diagnosed with COVID-19. In relation to demographic variables, the results obtained are consistent with the findings mentioned in the literature, namely that there is an association between COVID-19 with the male sex and the presence of comorbidities (such as diabetes and arterial hypertension) [23,24].

All patients required oxygen; however, in patients who received hormonal therapy, the need for respiratory support therapy, oxygen levels, and the need for mechanical ventilation and intubation significantly decreased, as did the days of the hospital stay. The use of hormonal therapy in COVID-19 patients also had a significant effect on some parameters of blood biometry, increasing the levels of erythrocytes and platelets. Differences were observed in the ESR, a decrease in the levels of CRP, and an increase in oxygen saturation.

Finally, in relation to the parameters associated with the outcome of the disease, it was observed that the patients who used hormonal therapy presented effects in terms of the levels of CRP, lymphocytes, D-dimer, and procalcitonin. The use of hormonal therapy has been shown to be effective against COVID-19. In an analysis of electronic health records, it was showed that the fatality risk for women > 50 years receiving estradiol therapy is reduced by more than 50% in comparison with the control [25]. It has been observed in an epidemiological comparative analyses of COVID-19 cases by country, sex, and age (in Australia, Columbia, Denmark, Italy, Mexico, Norway, Pakistan, Philippines, Portugal, Spain, Switzerland, and England) that the case fatality rate (CFR) of COVID-19 increases for both sexes with advancing age, but males have a significantly higher CFR than females at all ages greater than 30 years [11]. In this study, the authors suggest that biological sex differences contribute to male-biased death, but gender-associated risk of exposure may affect rates of infection differently for males and females, where the summative effect is a sex-specific transcriptional regulatory network of genetic variants, epigenetic modifications, transcription factors, and sex steroids that leads to a functional difference in the immune response [11]. Another study mentioned that the strong COVID-19 mortality associate with older age and with sex can be also related with other factors such as healthcare systems, patient characteristics, prevalence of diagnostic testing, type of comorbidities by sex, or the variability of populations [26]. Other studies support the idea that the survival advantage of women (under extreme conditions or epidemics) has fundamental biological underpinnings and also that the female advantage is modulated by a complex interaction of biological (e.g., sex differences in immune response), environmental (lifestyle), and social factors [27,28].

In an open-label randomized controlled trial, it was shown that the use of estradiol valerate (2 mg per day for 7 days) with the standard care in estrogen-deficient postmenopausal women infected with mild and moderate COVID-19 caused a significant decrease in the D-dimer, lactate dehydrogenase (LDH), interleukin (IL)-6, and CRP on day 5 of the intervention [29]. In addition to this study, it has been shown that the use of oral contraceptive pills in 18–45-year-old women infected with COVID-19 significantly reduced the hospital attendance and had a significantly lower predicted COVID-19 [30].

However, to our knowledge, the only existing study on sex hormones related to male patients with COVID-19 was a randomized, controlled pilot, open-label trial where it was observed that progesterone (100 mg subcutaneously twice daily for up 5 days) in addition to standard care in men hospitalized with moderate to severe COVID-19 reduced the number of days of oxygen supplementation and hospital stay compared to those who did not receive hormone therapy [31]. It has been observed that men are often affected by SARS-CoV-2; since they have a higher mortality rate when compared to women, the resistance to viral infection in women can be attributed to sex hormones, specifically estrogen, which is known to enhance the immune activity of both B as well as T-helper cells, and estrogens therefore act as an immune-stimulating factor [10,32]. Although the precise molecular mechanism is yet to be defined, it appears that estrogen competes with the spike’s receptor-binding domain (S-RBD) protein to bind specific sites that are used by the virus to bind the receptor, causing a reduction in energy on the surface of the receptor, rendering the receptor less susceptible to interact with other molecules, including those of SARS-CoV-2 [6].

Estrogen’s effects on innate immunity have been observed to include the suppression of pro-inflammatory cytokine production (IL-6, IL-1β and TNF-α) by monocytes and macrophages and the stimulation of antibody production by B cells [10]. Progesterone inhibits the pro-inflammatory cytokines IL-1β and IL-12 by macrophages and dendritic cells and promotes the production of the anti-inflammatory cytokines IL-4 and IL-10 [10,33]. Estrogens and progesterone also increase the expansion of regulatory T cells (Treg), promoting immunotolerance, and stimulate CD4+ T-helper cell production of anti-inflammatory cytokines [16]. Estrogens and progesterone have been proposed as possible candidates to mitigate the cytokine storm generated by SARS-CoV-2 while increasing antibody production. In the case of the present study, estrogen use was observed to decrease CRP and ESR levels. In clinical studies, it has been observed that ESR and CRP levels are directly related to the severity of COVID-19 [23,34,35]. The above observations demonstrate that men infected with COVID-19 and receiving this therapy are being benefited by the anti-inflammatory effects of estrogens, and this was observed based on a reduced need for adjuvant breathing therapy.

On the other hand, there is a relationship between sex hormones and the renin–angiotensin system. ACE2 expression is regulated by 17-β-estradiol in some organs such as the uterus, kidney, and heart. Estrogens have an effect on ACE2 in the heart and suppress RAS through the cleavage of an angiotensin II residue to release the angiotensin 1–7 metabolite, which possesses an antioxidant and anti-inflammatory effect [36], and it has been proposed that 17β-estradiol may reduce SARS-CoV-2 infection by diminishing elevated levels of two critical components, the ACE2 and transmembrane protease serine 2 (TMPRSS2), for SARS-CoV-2 cell entry [37]. Ethinylestradiol is used in formulations of contraceptives and hormone replacement therapy because it is an estradiol derivative; it is therefore conceivable that a similar mechanism to estradiol can be used to improve deleterious SARS-CoV-2 effects. Positive effects against SARS-CoV-2 in vitro have also been proposed in other female hormones, in particular progesterone, whose possible mechanism of action could be the increased accumulation of phospholipid, specifically the bis(monoacylglycero)phosphate, which, in synergy with chloroquine treatment, impairs the endosomal/lysosomal trafficking of SARS-CoV-2 and results in the virus being sequestered in multivesicular bodies [38].

The association of increased lymphocyte levels with poor prognosis in severe COVID-19 has also been observed [39], as well as increased levels of procalcitonin, serum ferritin, D-dimer, C-reactive protein, LDH, and cytokines, which have been used as inflammatory markers to monitor severity and progression in patients infected with COVID-19 [40]. Estrogen decreases the pro-inflammatory response in infected patients by decreasing the cytokine storm [16]. This is consistent with the results found in the present study, where the levels of procalcitonin, D-dimer, CRP, and lymphocytes decreased with the use of hormonal therapy in the patients studied and who survived the disease. The covariate adjustment analysis also showed that ferritin diminishes in the experimental group and that ESR does not increase. Ferritin, an iron-storing protein, is also known as a marker of severity in COVID-19 and has been correlated with severe COVID-19 [41], indicating that hyperferritinemia is caused by excessive inflammation during the disease due to inflammatory cytokines stimulating macrophages to secrete ferritin [42]. The ESR has been shown to be a prognostic factor for the disease severity and mortality in COVID-19 [43]. The ESR is an index of immunological loss. The use of hormonal therapy decreased the levels of acute-phase proteins, fibrinogen, and immunoglobulins related to the increase in ESR values in pulmonary disease [44].

However, as expected in a novel disease, limitations were present. The main one is the loss of subjects due to complications of COVID-19 in the second and third measurements, due to this fact some significant differences that could arise were hidden. We propose increasing the number of subjects and thus increasing the statistical power, which in the present study was 70%. Of note, this is the first study where transdermal hormonal patches were used as adjuvants in COVID-19 treatment in male patients. This route of administration is advantageous because it provides steady serum concentrations of drugs once weekly in a non-invasive fashion, is easy to acquire, is inexpensive, and is effective in preventing disease progression.

## 4. Materials and Methods

### 4.1. Trial Oversight

This pilot study was performed at the Centro Médico Nacional 20 de Noviembre and was a randomized, controlled trial. It was approved by the hospital Research and Ethics Committee (No. 07-203.2020) and did not receive any grants or private support. The trial was performed following the principles of the Declaration of Helsinki. The study design and collected data are publicly available at the clinical trial website with number NCT04539626. All authors assume full responsibility for the accuracy and completeness of the data and analyses, as well as for the fidelity of the trial and this report of our findings.

### 4.2. Patients

Hospitalized non-severe COVID-19 patients were considered for enrollment. We included patients of both sexes, men over the age of 18 and women over the age of 55, with RT-PCR-confirmed diagnosis. The recruitment was performed between November 2020 to March 2021. No participants were vaccinated at the moment of the recruitment. Participants were excluded when they presented recent hormonal treatment or estrogen-dependent cancer. All patients provided written informed consent; they had the possibility to abandon the study at any moment.

### 4.3. Trial Procedures

Participants were randomly assigned in a 1:1 ratio to either the experimental group or the control group. Randomization was performed with the use of a web-based system in permuted blocks, with block sizes ranging from 2 to 6 patients. Clinical follow-up of the participants was performed during their hospital stay, and clinical outcomes and biochemical parameters were registered in a database and were monitored at baseline, day 8, and day 15. The experimental group was treated with conventional COVID-19 protocol for hospitalized patients plus transdermal patches Evra^®^ (norelgestromin 6 mg and ethinyl estradiol 0.60 mg), which were applied on arrival to the hospital and were replaced at day 8 and day 15. The control group was treated only with the COVID-19 protocol, which, at the time of hospitalization, was formed by chloroquine (200 mg/kg) or azithromycin (500 mg/kg) or dexamethasone (8 mg/kg), depending on the progression of the international management of COVID-19.

### 4.4. Outcomes

The principal outcomes were the days of hospital stay, days of high-flow cannula oxygenation, days of mechanical ventilation, days of intubation, and oxygen saturation. The oxygen saturation was taken while patients were breathing without the cannula, in mechanical ventilation, and intubation, and the measurements were taken from the monitor. Additional outcomes were markers of inflammation, blood count, and clinical signs. The safety outcome of the experimental treatment was measured by hypercoagulation markers such as D dimer, fibrinogen, and platelets. Secondary and adverse effects were considered and were maintained under vigilance across the period of the study. Measurements were performed at hospital arrival, at day 8, and at day 15.

### 4.5. Statistical Analysis

The clinical trial was designed to have 80% statistical power to detect differences in the clinical and biochemical parameters between both groups. The Kolmogorov–Smirnov test was applied in order to determine the normality of the quantitative data. The group characteristics were compared with the use of Fisher’s exact test and Student’s *t*-test or Mann–Whitney U tests. Student’s *t*-test or Mann–Whitney U were employed to compare both groups’ baseline measurements. Next, we identified the outcomes that had statistical difference at baseline, in which case the covariate adjustment was performed for the analysis of final measurements between groups. Auto-controlled analysis was performed with the repeated measures ANOVA using Bonferroni test. Two-sided *p* values of 0.05 or less were considered to indicate statistical significance. Analyses were performed with GraphPad Prism software, version 8.0.0 for Windows (GraphPad Software, San Diego, CA, USA), and SPSS software, version 23 (IBM Corp. Released 2015. IBM SPSS Statistics for Windows, Version 23.0. Armonk, NY, USA: IBM Corp).

## 5. Conclusions

Norelgestromin and ethinyl estradiol transdermal patches reduced the days of hospital stay, the days of high flow oxygen supplementation, the days of mechanical ventilation, and the days of intubation and increased the oxygen saturation. Whereas ferritin significantly decreased and ESR did not increase in comparison with the control group. Transdermal patches did not impact coagulation markers. Thus, this could be considered a viable treatment for COVID-19 patients with moderate symptoms.

## Figures and Tables

**Figure 1 pharmaceuticals-15-00757-f001:**
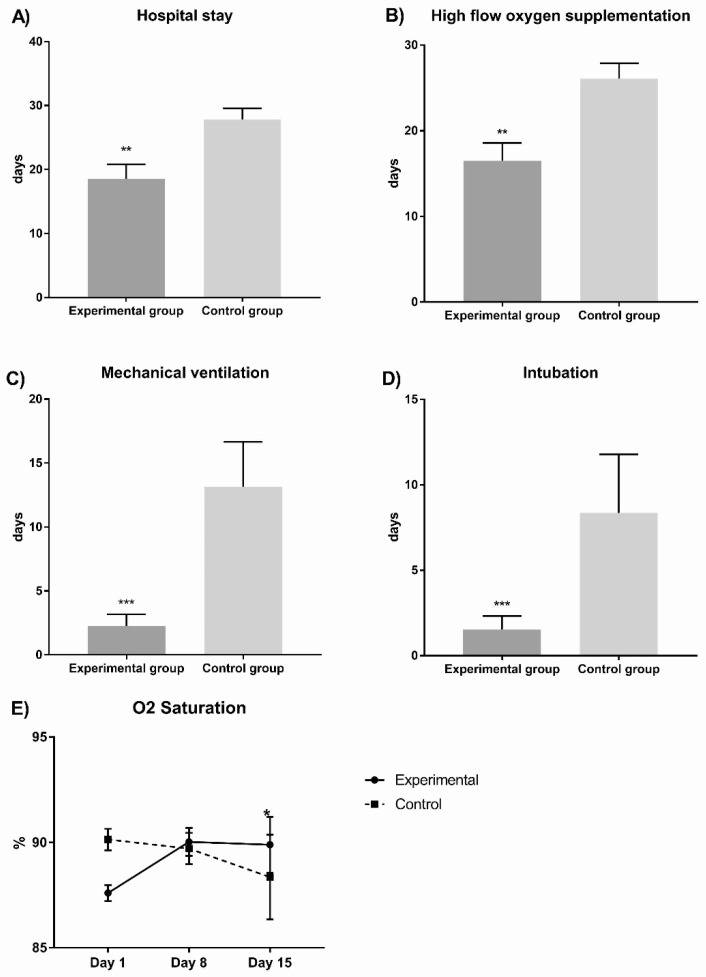
Comparisons between experimental and control groups of principal outcomes evaluated in COVID-19 patients. In (**A**) we observed the hospital stay; in (**B**) we observed the high flow oxygen supplementation; in (**C**) we observed the mechanical ventilation; in (**D**) we observed the intubation and in (**E**) we observed the oxygen saturation during hospital stay days comparing both experimental vs control group. The experimental group presented significantly fewer hospital stay days (*p* = 0.01), high-flow-oxygen-supplementation days (*p* = 0.001), mechanical-ventilation days (*p* = 0.003), and intubated days (*p* = 0.01); in contrast, oxygen saturation significantly increased (*p* = 0.01 * *p* < 0.05, ** *p* < 0.01, *** *p* < 0.001.

**Figure 2 pharmaceuticals-15-00757-f002:**
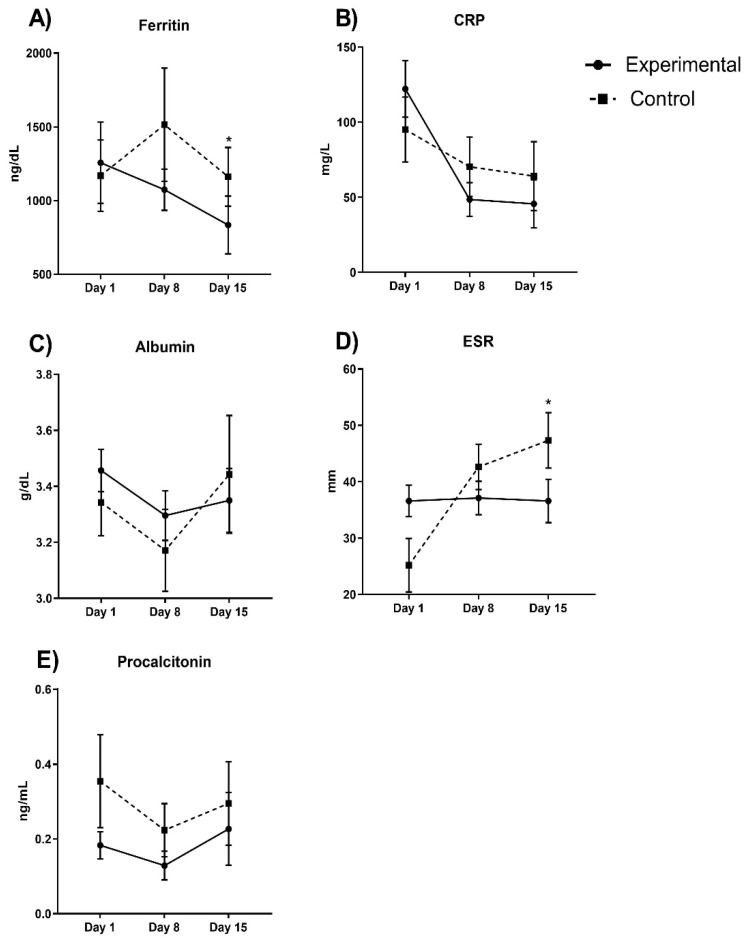
Inflammatory parameters between experimental and control groups evaluated in COVID-19 patients. Ferritin diminishes in the experimental group approximately by 50% (*p* = 0.001). In (**A**) we observe ferritin values; (**B**) CRP levels; in (**C**) albumin; (**D**) ESR and (**E**) procalcitonin during hospital stay days comparing both experimental vs control group. Auto-controlled analysis did not show significant changes in the experimental in ESR (*p* > 0.05), but in the control group, it significantly increased (*p* = 0.0001). Considering a covariate adjustment, experimental group ESR was significantly decreased in comparison with the control group (*p* = 0.01). CRP and procalcitonin did not significantly decrease in the experimental group (74% and 33%, respectively; *p* > 0.05 in both). * *p* < 0.05.

**Figure 3 pharmaceuticals-15-00757-f003:**
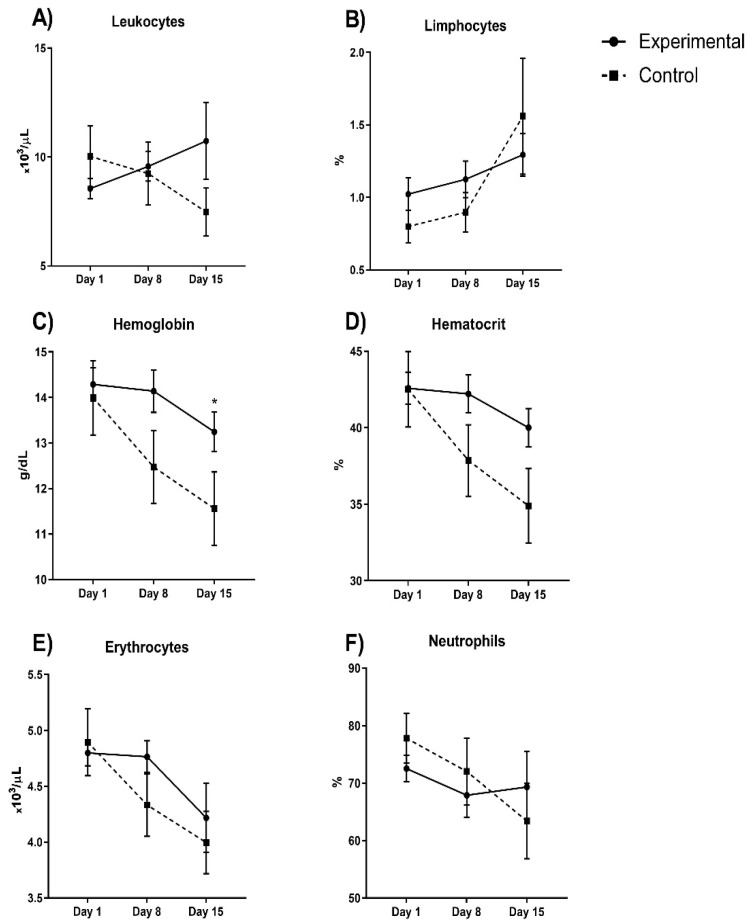
Blood count parameters between experimental and control group evaluated in COVID-19 patients (**A**) leukocytes; (**B**) lymphocytes; (**C**) hemoglobin; (**D**) hematocrit; (**E**) erythrocytes and (**F**) neutrophils during hospital stay days comparing both experimental vs control group. Hemoglobin and hematocrit decreased in the experimental group by 5.6 and 9.7%, respectively (*p* > 0.05). Cellular counting shows that erythrocyte and neutrophils decreased (44% and 10%, respectively), but not significantly, in the experimental group while leukocytes and lymphocytes increased, but not significantly (10.2% and 34.3%, respectively). * *p* < 0.05.

**Table 1 pharmaceuticals-15-00757-t001:** Clinical and demographic data of the subjects.

	Experimental Group (n = 30)	Control Group (n = 14)	*p* Value
Age (mean ± SE)	55.13 ± 2.5	54.5 ± 6.01	0.91
Sex (n, women/men)	12/18	1/13	0.02
Overweight/obesity (n)	2, 6.7%	1, 7.1%	0.69
Diabetes mellitus (n)	11, 36.7%	4, 28.6%	0.43
Arterial hypertension (n)	8, 36.7%	5, 35.7%	0.39
Azithromycin 500 mg/kg (n)	27, 90.0%	14, 100%	0.30
Chloroquine 200 mg/kg (n)	19, 63.3%	10, 71.4%	0.43
Dexamethasone 8 mg/kg (n)	19, 63.3%	10, 71.4%	0.43

SE = Standard error.

**Table 2 pharmaceuticals-15-00757-t002:** Clinical signs patients and safety outcomes measured of COVID-19.

	Experimental Group (n = 30)			Control Group (n = 14)		
Clinical signs (n, %)	*Day 1*	*Day 15*	*p value*	*Day 1*	*Day 15*	*p value*
Fever	26 (86.67%)	4 (13.79%)	0.0001	14 (100%)	2 (14.29%)	0.0001
Cough	30 (100%)	22 (75.86%)	0.005	14 (100%)	8 (57.14%)	0.016
Dyspnea	26 (86.67%)	6 (20.69%)	0.0001	13 (92.86%)	2 (14.29%)	0.0001
Diarrhea	14 (46.67%)	3 (10.34%)	0.0001	9 (64.29%)	3 (21.43%)	0.02
ARDS	28 (93.33%)	5 (16.67%)	0.0001	11 (78.57%)	4 (28.57%)	0.008
Safety outcomes (mean ± SD)	*Day 1*	*Day 15*	*p value*	*Day 1*	*Day 15*	*p value*
D dimer ng/dL	5.34 ± 2.57	5.58 ± 2.18	1.0	6.49 ± 2.66	4.21 ± 2.26	1.0
Fibrinogen mg/dL	737.06 ± 220.9	569.46 ± 116.8	0.05	557.3 ± 204.05	482.27 ± 168.7	0.89
Platelets ×10^3^/µL	300.56 ± 25.8	314.43 ± 36.16	1.0	228.78 ± 27.68	239.42 ± 38.65	1.0

ARDS = acute respiratory distress syndrome; SD = standard deviation.

## Data Availability

The study design and collected data are publicly available at the clinical trial website with number NCT04539626.

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
