# Peer review of "Effect of Norelgestromin and Ethinylestradiol in Transdermal Patches on the Clinical Outcomes and Biochemical Parameters of COVID-19 Patients: A Clinical Trial Pilot Study"

_pharmaceuticals, 2022, doi:10.3390/ph15060757_

Round 1

Reviewer 1 Report

There are many pieces of evidence that estrogens have an effect on COVID-19 mortality where women of fertile age seem to be protected against the serious course of the disease. This study demonstrated the effect of transdermal patches loaded with hormones on the progress of the disease, where this therapy reduced hospital stay, oxygenation, mechanical ventilation, and intubation, and improved patient oxygenation. These data are encouraging for further therapy of COVID-19 patients. The article is well written but I believe, that is necessary to support him with additional references that will explain the potential molecular effect of hormones on COVID-19 infection. 

Author Response

Response to Reviewer 1 Comments

Point 1: There are many pieces of evidence that estrogens have an effect on COVID-19 mortality where women of fertile age seem to be protected against the serious course of the disease. This study demonstrated the effect of transdermal patches loaded with hormones on the progress of the disease, where this therapy reduced hospital stay, oxygenation, mechanical ventilation, and intubation, and improved patient oxygenation. These data are encouraging for further therapy of COVID-19 patients. The article is well written but I believe, that is necessary to support him with additional references that will explain the potential molecular effect of hormones on COVID-19 infection. 

Response 1: Dear reviewer, thank you for your valuable comments. Following your suggestions, we added information that supports and explains the potential molecular effect of estrogen and progesterone in the context of COVID-19 infection.

Reviewer 2 Report

  The authors are to be congratulated on their work presenting a novel approach to an important clinical problem and it shows promising results.

  The presentation can be improved in several areas.

  In the introduction, the section on ACE2 is not directly relevant to the work presented since hormonal therapies do not directly impact the entry of the virus into cells, and I would omit it. The authors suggest that the hormones may play a role in disease onset without a relevant citation and their treatment involves patients with established disease; it is not clear this statement is relevant or supported by prior publications so I would consider omitting it.

  In the presentation of the results, the authors refer to "oxygenation days". It is not clear to me what this means. Are they referring to high flow oxygen supplementation or any use of supplemental oxygen? This should be clarified. They present a figure showing changes in oxygen saturation but do not provide information regarding use of supplemental oxygen; the information should be clarified to indicate if there were patients using supplemental oxygen or if these measurements were made while patients were breathing room air. In describing figure 3, the authors use the term "biometry"; this is not a standard term and should be changed to clarify. "Neutrophils" is misspelled as "neutrophiles". The first sentence of section 3.1.3 is unclear and should be fixed.

  In the discussion, the authors point out that older age is associated with greater mortality from COVID-19 but that even among older patients, the mortality is less in women; this would be an argument against their hypothesis that sex hormones are an important factor contributing to the improved survival of women since older women produce less of these hormones compared to younger women. In the section "It has been observed that women generally develop a 181 better immune response compared to men, being the difference in concentration of hor-182 mones one of the most important factors that regulate the immune response, since sex 183 steroids are potent immune modulators [13]. In relation to the above, it has been observed 184 that serum estrogen and progesterone concentrations are related to the severity of SARS-185 CoV-2 infection since they decrease the formation of pro-inflammatory cytokines [25]." the language is awkward preventing this section from making the authors' point adequately. Near the end of the discussion, the authors talk about the relationship between ferritin levels and severe COVID-19; there appears to be a typographical error in the sentence "Ferritin, an iron-storing protein, also is known as a marker of severity in COVID-19 and has been correlated with severe COVID-1935 indicating that hyperferritinemia is caused by excessive inflammation during the disease due to inflammatory cytokines stimulating to macrophage to secrete ferritin [36]." It appears that the number 1935 should in stead indicate COVID-19 and the 35 should be a reference. At the end of the discussion, the authors refer to "this via of administration". This is not standard English and should be corrected.

  In the Materials and Methods section, the authors describe their inclusion criteria "We included patients of both sexes, older than 18 years and women older than 55 years, with RT-PCR confirm diagnosis." It appears that they included men over the age of 18 and women over the age of 55; if this is correct, the sentence should be clarified. In section 4.4, describing the outcomes evaluated, the first sentence should be clarified.

Author Response

Response to Reviewer 2 Comments

Point 1: The authors are to be congratulated on their work presenting a novel approach to an important clinical problem and it shows promising results. The presentation can be improved in several areas.

Response 1: We are grateful for your valuable comments.

Point 2: In the introduction, the section on ACE2 is not directly relevant to the work presented since hormonal therapies do not directly impact the entry of the virus into cells, and I would omit it. The authors suggest that the hormones may play a role in disease onset without a relevant citation and their treatment involves patients with established disease; it is not clear this statement is relevant or supported by prior publications so I would consider omitting it.

Response 2: We added references that explain better the molecular effect of estrogen and progesterone in COVID-19. We consider that it is important to mention, that angiotensin 1-7 (ang 1-7) has a cytoprotective effect, and ang 1-7 could be decreased because it is synthetized from the metabolism of angiotensin II by ACE2, which is the virus receptor, thus, decreasing its activity due to its interaction with viral proteins. (https://www.frontiersin.org/articles/10.3389/fimmu.2021.658428/full) The increase in angiotensin II generates cell damage by the action of the ATR1 receptor (hyperinflammation and oxidative stress damage). It has been shown in other diseases, that estrogenic derivatives can increase ang 1-7, which would counteract damage, by acting on the MAS receptor, reducing inflammation and damage by free radicals. We consider that in COVID-19, estrogen derivatives and progesterone could not only act beneficially through the cellular immune response but also by increasing ang 1-7 (https://www.sciencedirect.com/science/article/abs/pii/S0303720715301295). But we still must corroborate this hypothetical idea, so we focus the introduction only to improve the foundation that you point out to us.

Point 3: In the presentation of the results, the authors refer to "oxygenation days". It is not clear to me what this means. Are they referring to high flow oxygen supplementation or any use of supplemental oxygen? This should be clarified.

Response 3: We are referring to the days those patients required high flow oxygen supplementation. We clarified this point into the new article´s version.

Point 4: They present a figure showing changes in oxygen saturation but do not provide information regarding use of supplemental oxygen; the information should be clarified to indicate if there were patients using supplemental oxygen or if these measurements were made while patients were breathing room air.

Response 4: The oxygen saturation was taken while patients were breathing without the cannula. We describe this in the material and methods.

Point 5: In describing figure 3, the authors use the term "biometry"; this is not a standard term and should be changed to clarify.

Response 5: We replace the word “biometry” by “blood count” in figure 3, results and in material and methods.

Point 6:  "Neutrophils" is misspelled as "neutrophiles".

Response 6: We replace the wrong word "neutrophiles" by "Neutrophils" in figure 3 and results.

Point 7: The first sentence of section 3.1.3 is unclear and should be fixed.

Response 7: We have modified the wording of this section to make the information clearer, as rightly suggested.

 Point 8: In the discussion, the authors point out that older age is associated with greater mortality from COVID-19 but that even among older patients, the mortality is less in women; this would be an argument against their hypothesis that sex hormones are an important factor contributing to the improved survival of women since older women produce less of these hormones compared to younger women. In the section "It has been observed that women generally develop a 181 better immune response compared to men, being the difference in concentration of hor-182 mones one of the most important factors that regulate the immune response, since sex 183 steroids are potent immune modulators [13]. In relation to the above, it has been observed 184 that serum estrogen and progesterone concentrations are related to the severity of SARS-185 CoV-2 infection since they decrease the formation of pro-inflammatory cytokines [25]." the language is awkward preventing this section from making the authors' point adequately.

Response 8: We modify the writing as suggested to clarify the idea.

Point 9: Near the end of the discussion, the authors talk about the relationship between ferritin levels and severe COVID-19; there appears to be a typographical error in the sentence "Ferritin, an iron-storing protein, also is known as a marker of severity in COVID-19 and has been correlated with severe COVID-1935 indicating that hyperferritinemia is caused by excessive inflammation during the disease due to inflammatory cytokines stimulating to macrophage to secrete ferritin [36]." It appears that the number 1935 should in stead indicate COVID-19 and the 35 should be a reference.

Response 9: We corrected the mistake that correctly pointed out to us.

Point 10: At the end of the discussion, the authors refer to "this via of administration". This is not standard English and should be corrected.

Response 10: We replaced "this via of administration” by route of administration.

Point 11: In the Materials and Methods section, the authors describe their inclusion criteria "We included patients of both sexes, older than 18 years and women older than 55 years, with RT-PCR confirm diagnosis." It appears that they included men over the age of 18 and women over the age of 55; if this is correct, the sentence should be clarified.

Response 11: We rewrite this statement for better understanding. We included men older than 18 years and women older than 55.

Point 12: In section 4.4, describing the outcomes evaluated, the first sentence should be clarified.

Response 12: We improved the wording and changed terms to clarify the idea we want to convey.

Reviewer 3 Report

The manuscript written by Alfredo presents interesting research. However, it needs to address several issues.

The title needs to be rephrased, evolution is misleading, the study was merely done for 15 days… changes or something else will fit..

The very first sentence uses pandemic two times, it needs to correct.

Need clarification on the experimental group. Did they received any treatment apart from TP? authors say that ‘Control followed the conventional COVID-19 treatment protocol’. What do you mean? Is this conventional treatment was denied for experimental group?

Unfortunately, there seems to be incomplete introduction. They were discussing the mechanism of action of hormones on immunity, but they did not build on that. They need to elaborate and introduce the topic with the help of literature and the gaps present in them.

The gender distribution is not equal, significant difference in the number of men and women in the control group, this will direct impact on outcome analysis.

Did all patients put under mechanical ventilation? How was it calculated? Even intubation days, how?

Was oxygen saturation average of all recordings or given for every time of measurement?

How many patients were suffering from severe and how many moderate? What was the criteria for their classification? Who classified them?

As authors cited a study that has shown better response in women with hormone therapy in COVID-19 than men. In this study, in control group only one women was included while 13 were men, it is possible that the beneficial effect seen in experimental group could be probably due to being female high number than in control (since average is taken with high females compared to control group).  Authors need to clarify.

What about the reliability of the study where you had less than 50% samples in the control compared to experimental group?

If the participants were assigned in 1:1 ratio in two group, why the number decreased in control group. It would be better to include a chart about the sampling steps.

How was the procedure of informed consent done? Was the patient involved or their family members? Which language was used for conveying the information of the study?

Author Response

Response to Reviewer 3 Comments

Point 1: The manuscript written by Alfredo presents interesting research. However, it needs to address several issues.

Response 1: Dear reviewer.  We are grateful for your valuable comments.

Point 2: The title needs to be rephrased, evolution is misleading, the study was merely done for 15 days… changes or something else will fit.

Response 2: We modify the title according to your suggestion.

Point 3: The very first sentence uses pandemic two times, it needs to correct.

Response 3: We performed the suggested modification.

Point 4: Need clarification on the experimental group. Did they received any treatment apart from TP? authors say that ‘Control followed the conventional COVID-19 treatment protocol’. What do you mean? Is this conventional treatment was denied for experimental group?

Response 4: Both groups received the same treatment protocol, however, the subjects in the Experimental Group received an additional treatment with transdermal patches. We clarify this misunderstand in the pertinent section.

Point 5: Unfortunately, there seems to be incomplete introduction. They were discussing the mechanism of action of hormones on immunity, but they did not build on that. They need to elaborate and introduce the topic with the help of literature and the gaps present in them. 

Response 5: Dear reviewer, we really appreciated your comments. As suggested, we increased the information regarding the mechanisms of action of estrogens and progesterone in relation to immunity and the fact that there are reports (https://www.sciencedirect.com/science/article/abs/pii/S0303720715301295) as well as in other pathologies it has been shown that estrogen derivatives can increase ang 1-7 (https://journals.physiology.org/doi/full/10.1152/ajpregu.00182.2014); this in COVID-19 is extremely important in cell cytoprotection (https://pubmed.ncbi.nlm.nih.gov/24193198/). In COVID-19 severe patients circulating levels of ang 1-7 was found to be decreased (https://onlinelibrary.wiley.com/doi/full/10.1002/hsr2.564).  In this sense, clinical trials are being carried out with ang 1-7 agonists for patients with severe COVID-19 (https://www.ncbi.nlm.nih.gov/pmc/articles/PMC7539308/), however, it has been seen that it is easily degraded when it is injected intravenously in patients. Then, other way to increase ang 1-7 could be endogenous through patches with estrogen derivatives making this study promising, adding to the immunomodulatory benefits of estrogen and progesterone (https://academic.oup.com/endo/article/161/9/bqaa127/5879027). 

Point 6: The gender distribution is not equal, significant difference in the number of men and women in the control group, this will direct impact on outcome analysis.

Response 6: In all the other variables, significant differences were not found, plus we consider confounding variables when performing the covariable adjustment. Thus, this will not significantly impact on the outcomes.

Point 7: Did all patients put under mechanical ventilation? How was it calculated? Even intubation days, how? Was oxygen saturation average of all recordings or given for every time of measurement?

Response 7: Not all patients required mechanical ventilation. The days in wich the patients were under mechanical ventilation were defined by the pneumologist. Mechanical ventilation was different from intubation. The recordings of oxygen saturation presented in the figures was the average. All decisions of the clinical management were taken by the pneumologist according to the patient clinical evolution.

Point 8: How many patients were suffering from severe and how many moderate? What was the criteria for their classification? Who classified them?

Response 8: At the day of hospitalization, all patients were classified as moderate by the emergency physician.

Point 9: As authors cited a study that has shown better response in women with hormone therapy in COVID-19 than men. In this study, in control group only one women was included while 13 were men, it is possible that the beneficial effect seen in experimental group could be probably due to being female high number than in control (since average is taken with high females compared to control group).  Authors need to clarify.

Response 9: There was a greater proportion of female in the experimental group, but as was mentioned above, all the other confounding variables were taken into account when performing the covariate adjustment, reducing the possible effect that gender could add to our results. Plus, this is a pilot study, and the number of participants was limited.

Point 10: What about the reliability of the study where you had less than 50% samples in the control compared to experimental group?

Response 10: The statistical design that we employed was with the purpose of decreasing bias that resulted from the difference in subjects in each group.

Point 11: If the participants were assigned in 1:1 ratio in two group, why the number decreased in control group. It would be better to include a chart about the sampling steps.

Response 11: Considering the context of the pandemic, not all the patients that arrived to the hospital qualified for inclusion and exclusion criteria, plus the emotional state of the patients and the family was not always the same, resulting in a negativity on hospitalization; all of this factors complicated the recruitment in the current study.

Point 12: How was the procedure of informed consent done? Was the patient involved or their family members? Which language was used for conveying the information of the study?

Response 12: At the time of hospitalization all patients were considered moderately and mentally conscious, allowing self-decision for the participation on the trial, plus the family members were informed and signed as witnesses in the informed consent. The language employed was colloquial in terms of explaining the objectives and possible outcomes to the participants and their family.

Round 2

Reviewer 3 Report

Since authors did substantial changes in the manuscript based on reviewers recommendation and the present version is improved form, I recommend for acceptance.